# Understanding the Role of Epithelial Cells in the Pathogenesis of Systemic Sclerosis

**DOI:** 10.3390/cells14130962

**Published:** 2025-06-24

**Authors:** Lydia Nagib, Anshul Sheel Kumar, Richard Stratton

**Affiliations:** Centre for Rheumatology and Connective Tissue Diseases, UCL Division of Medicine, London NW32PF, UK; lydiagadalla.nagib@nhs.net (L.N.); anshulsheelkumar@gmail.com (A.S.K.)

**Keywords:** systemic sclerosis, epidermis, fibrosis, fibroblast, EMT, ECM

## Abstract

Systemic sclerosis (SSc) is an autoimmune fibrotic disorder affecting the skin and internal organs, categorized as either limited cutaneous SSc, where distal areas of skin are involved, or diffuse cutaneous SSc, where more extensive proximal skin involvement is seen. Vascular remodelling and internal organ involvement are frequent complications in both subsets. Multiple pathogenic mechanisms have been demonstrated, including production of disease-specific autoantibodies, endothelial cell damage at an early stage, infiltration of involved tissues by immune cells, as well as environmental factors triggering the onset such as solvents and viruses. Although not strongly familial, susceptibility to SSc is associated with multiple single nucleotide polymorphisms in immunoregulatory genes relevant to antigen presentation, T cell signalling and adaptive immunity, as well as innate immunity. In addition, several lines of evidence demonstrate abnormalities within the epithelial cell layer in SSc. Macroscopically, the SSc epidermis is pigmented, thickened and stiff and strongly promotes myofibroblasts in co-culture. Moreover, multiple activating factors and pathways have been implicated in the disease epidermis, including wound healing responses, induction of damage associated molecular patterns (DAMPS) and the release of pro-fibrotic growth factors and cytokines. Similar to SSc, data from studies of cutaneous wound healing indicate a major role for epidermal keratinocytes in regulating local fibroblast responses during repair of the wound defect. Since the epithelium is strongly exposed to environmental factors and richly populated with protective immune cells, it is possible that disease-initiating mechanisms in SSc involve dysregulated immunity and tissue repair within this cell layer. Treatments designed to restore epithelial homeostasis or else disrupt epithelial–fibroblast cross-talk could be of benefit in this severe and resistant disease. Accordingly, single cell analysis has confirmed an active signature in SSc keratinocytes, which was partially reversed following a period of JAK inhibitor therapy.

## 1. Introduction

Systemic sclerosis (SSc) is a heterogenous, chronic, multisystem disorder, in which dysregulated immunity and vascular dysfunction are associated with progressive fibrosis, which spreads in a continuous fashion through the affected skin and frequently involves internal organs such as the lungs, leading to impaired organ function and reduced life-expectancy [1]. How these various pathogenic elements are being initiated and interconnected is not fully understood. However, it is believed that there is an underlying interplay between immune cell activation, both innate and adaptive, and the induction of fibroblasts, leading to persistence of myofibroblasts responsible for skin and organ fibrosis [2]. Endothelial cell damage is usually present at the earliest clinical stages, evidenced by clinically visible microvascular damage, enhanced adhesion molecule expression and loss of protective endogenous vasodilator regulation, leading to immune cell adhesion and transmigration, plus vasoconstriction and associated tissue ischaemia and Raynaud’s phenomenon [3]. In the resulting inflammatory and ischaemic microenvironment, there is excess synthesis of extracellular matrix (ECM) proteins, mainly collagens I and III, which elevate stiffness and mechanical stress within the local tissue and displace normal physiologic tissue-resident cells. Activation of myofibroblasts is essential to normal wound healing, where it is coordinated and self-limiting, whereas in SSc, there is persistence of the activated state of the myofibroblasts due to the formation of the stiffened and growth factor-enhanced micro-environment, leading to a feed-forward loop [4]. Single cell analysis has given insights into the myofibroblast heterogeneity in the skin, indicating the presence of COMP+, COL11A1+, SFRP4/SFRP2+ and PRSS23/SFRP2+ fibroblast populations, as well as proinflammatory CCL19+ fibroblasts, each of which were elevated in proportion to clinical skin severity [5]. Other fibroblast clusters were found to be reduced, specifically CXCL12-expressing and PI16-positive groups.

Whilst there has been a very extensive and high-level investigation into immune abnormalities, the role of endothelial dysfunction and fibroblast abnormalities underlying SSc, the potential contribution of the epithelial cell layer has received less attention. In the fibrotic skin lesions, the pathogenic myofibroblasts are arising from multiple sources, including local tissue-resident fibroblasts, infiltrating monocyte-derived cells (“fibrocytes”) [6], subcutaneous fat-derived mesenchymal stem cells (MSCs) [7,8], perivascular stem cells (pericytes) [9] and transdifferentiating tissue-resident cells, such as endothelial cells undergoing endothelial to mesenchymal transition (endoMT) [10], but also a potential role for epithelial cells undergoing partial epithelial to mesenchymal transition (EMT). Variation in the relative contribution of myofibroblast precursors from different sources could explain the heterogeneity of this disease and is of potential relevance to the myofibroblast subpopulations now identified through single cell analysis.

Accordingly, several lines of evidence support the notion that dysregulated epithelial cells are contributing to the fibrotic process of SSc (summarised in Table 1) [11,12,13,14,15,16]. Activated epithelial cells are already known to be important in other fibrotic diseases, such as idiopathic lung fibrosis [17,18] and renal fibrosis [19]. The pathogenesis of idiopathic pulmonary fibrosis (IPF) has undergone significant re-evaluation over recent decades. Traditionally, IPF, similar to many interstitial lung diseases, was considered to be driven by a chronic, unresolved interstitial inflammatory process triggered by an unknown insult [20]. However, IPF is now increasingly viewed as a disease primarily driven by aberrant wound healing triggered by dysregulated epithelial–mesenchymal interactions following repeated, subclinical alveolar epithelial injury [21].

It is also well-established that in the restoration of tissue homeostasis after wound healing, the cross-talk between epithelial cells and fibroblasts has an important role to play [22]. In this review, we will aim to discuss the potential contribution of the dysregulated epithelial cell layer in the pathogenesis of SSc.

**Table 1 cells-14-00962-t001:** Summary of phenotypic changes in the SSc epidermis.

Phenotype Change	Molecular Basis	Reference
Delayed differentiation of basal keratinocytes.	Continued expression of K5 and 14 into suprabasal and spinous layers.	[12]
EMT	Induction of SNAI and SFRP4. Altered Wnt signalling.	[11,23,24]
Wound healing phenotype.	Expression of K6 and K16 usually seen in wound epidermis.	[13,15,16]
Thickening of epidermis.	Increased number of keratinocyte cell layers. Increase size of keratinocytes.	[14]
Hyperproliferation.	Ki67 staining shows increased basal keratinocyte proliferation.	[14]
Altered gene expression signature.	Type I IFN signature responsive to JAK/STAT inhibitor.	[25]
Hyperpigmentation.	Increased melanocyte content. Altered CCN3 expression by melanocytes. Associations with early diffuse subset and race.	[26,27,28]
Altered barrier function.	Expansion of involucrin and loricrin-positive cell layers.	[14]
Pro-fibrotic activity	Activation of dermal fibroblasts in 3D co-culture. Media transfer activation of fibroblasts.	[13,16,29]

## 2. Altered Differentiation of the Epidermis in Scleroderma: Activated Wound Healing Phenotype

The epidermal cell layer represents a specialised stratified squamous epithelium capable of coordinated self-repair (reviewed in [30]). Keratinocytes have an intrinsic up–down polarity, and alter function according to local micro-environment cues, such as cell–cell contact, cell–ECM adhesion, mechano-sensing and metabolic factors such as the local partial pressure of oxygen and acid–base balance [31]. In health, basal keratinocytes express high levels of the cytokeratins 5 and 14 (K5 and K14), whereas these factors are lost as the keratinocytes commit to differentiation and migrate upwards, switching to cytokeratins 1 and 10 (K1 and K10) [32], a process which is markedly altered in SSc, where basal K14 positive cells abnormally persist into the spinous and granular layers, whereas induction of K10 is delayed [11,15]. Furthermore, involucrin and loricrin, proteins which are involved in the formation of the keratinocyte envelope, are expressed in an expanded distribution in the SSc epidermis [13,16]. These findings are seen consistently across a range of SSc biopsies, accompanied by increased keratinocyte cell size and expanded epidermal thickness [13] (summarised in Figure 1). These changes resemble those seen during wound healing, where the epidermis thickens and keratinocytes switch to alternative cytokeratins K6 and K16, which are involved in the adaptation to the wound environment [33].

Following injury, the epidermis is involved in a coordinated wound-healing process that evolves over stages categorised as: homeostasis, inflammation, proliferation and remodelling [34]. Injury-induced signatures can persist in the keratinocyte layer and are also known to spread laterally; for example, after wounding. Immediately after skin injury, there is vascular constriction and fibrin clot formation. At this stage, disruption of the epidermal barrier causes the release of pre-stored IL-1α by keratinocytes. Platelets degranulate into the damaged tissue, releasing platelet-derived growth factor (PDGF), transforming growth factor beta (TFGβ) and epidermal growth factor (EGF). The formed clot provides a matrix for inflammatory cells to infiltrate the wound. IL-8 released downstream of Toll-like receptor (TLR) signalling and complement factor 3 fragments (C3a) act as chemokines for neutrophils, which are recruited into the wound and remove bacteria from the wound site. TFGβ, which is released in the wound by the action of thrombin on the tissue-resident latent form of this growth factor, promotes differentiation of infiltrating monocytes into macrophages. In the early inflammatory phase, there is predominant M1 polarity, with macrophages releasing pro-inflammatory interleukin (IL)-1β and 1L-6, followed by a switch to M2 phenotype, which is more adapted to promoting fibroblast responses, enhancing local collagen ECM synthesis and driving neo-angiogenesis [35]. In the proliferative phase, or re-epithelialisation, keratinocytes become activated and begin to express keratins K6 and K16, which facilitate keratinocyte migration and enable the cells to withstand wound environments and cover wound defects [36]. Accordingly, in SSc, abnormal expression of K6 and K16 in the uninjured epidermis has been demonstrated [12] and confirmed by other studies [15,16], consistent with the notion that wound healing mechanisms are being induced in this cell layer.

Current data do not fully explain the altered keratinocyte phenotype in SSc. Profiling has demonstrated altered homeobox gene regulation, suggesting fundamentally altered imprinting [16]. Pigmentary changes are also evident in SSc, indicating enhanced delivery of pigmentary granules into the basal keratinocytes. Studies have attempted to explain and link these pigmentary changes to underlying pathogenic mechanisms in SSc [26]. The altered pigmentation is apparent in around 50 percent of SSc cases, broadly categorised as being of two patterns: (1) vitiligo-like, associated with perifollicular hyperpigmentation also termed “salt and pepper changes”, or (2) a more diffuse hyperpigmentation phenotype. The latter phenotype is typically seen in severe diffuse SSc patients with higher modified Rodnan skin scores, and also with a higher frequency of digital ulceration. Of relevance, absolute numbers of melanocytes are elevated in early-stage SSc patients (<5 years duration) and then subsequently fall in number with late-stage SSc [34]. CCN3 (also called NOV), a regulatory matricellular protein of reduced activity in fibrosis, is found to decrease in melanocytes in SSc as a potential contributory change [27]. The racial background is also important, with SSc-associated pigmentary changes more prevalent in non-Caucasians [28]. It is plausible that the elevated melanocyte content in SSc lesional skin is induced by enhanced chemokine recruitment of melanocytes or else by increased survival in the SSc epithelial layer. Increased stem cell factor (SCF, Kit-ligand) in SSc epidermis could potentially recruit melanocytes since they are known to express the receptor c-Kit [37].

## 3. The Epidermis as a Potential Site for Initiating Autoimmunity in SSc

The essential role of the epidermis in innate and adaptive immunity is firmly established based on multiple publications and has been reviewed [38]. This cell layer represents a major innate immune barrier to invading pathogens; moreover, it is a site where highly specific adaptive immune responses can be initiated, as seen in the use of intra-epidermal vaccines [39]. Overall, the epidermis has an important immunologic function, being the first barrier of defence against invading organisms and has its own dedicated population of innate and adaptive immune cells. Langerhans cells (LCs), previously believed to be a type of dendritic cell but now shown to align with a specially adapted type of macrophage, are resident in the basal and suprabasal epidermis and, upon activation, migrate away to the draining lymph nodes [40]. These cells are considered essential initiators of immune responses through their role in antigen processing and presentation to effector T cells. LCs are identified by expression of lanerin (CD207) and by the presence of Birbeck granules. Moreover, as immune regulators, they can determine immune tolerance or the breaking of immune tolerance in the early stages of autoimmunity [41] and can interact directly with T cells in the normal epidermis [42]. Although potentially important, LCs have not been fully explored in SSc with current methods; however, previous published data have indicated reduced LCs in the epidermis [43] through antibody staining, confirmed in a further study [44]. One possibility is that following local immune activation, LCs migrate away to draining lymph nodes. Functional studies could be achieved through sampling and culture of the epidermis sheet in SSc, which can routinely be separated by the suction blister method or else detached from punch biopsies.

In fact, various T cell subsets are known to reside in the skin, exceeding those present in the circulation in total number and categorised as γδ T cells and αβ T cells, the latter making up the great majority [45]. Of the αβ T cells, these are mainly present as the memory T cell phenotype, indicating expansion in response to previous antigen exposure and characterised as resident memory T cells (T_RM_ cells) [46]. In human skin, the T_RM_ are subdivided into CD103+ and CD103–, of which the CD103+ T cells are mainly in the epidermis, whilst CD103− T_RM_ are in the dermis. These populations can change rapidly following infective or other environmental insults and are responsive to changes such as cellular activation in the tissue resident cells, including keratinocytes [47]. Hence, dysregulated activation of the keratinocyte cell layer could form an initiating substrate for antigen-specific T cell responses contributing to the SSc disease process. As alluded to above, explants of SSc epidermal tissue could be utilised to model and profile effector T cell responses, including the T cell receptor repertoire and their clonal expansion.

The functions of immune cells and endothelial cells can be influenced by exosomes released by keratinocytes, as these cells are capable of secreting chemo-attractants, anti-microbials and cytokines, as well as DAMPs. The functions of keratinocytes as the first line of defence are mediated by the expression of receptors for DAMPS and PAMPS, including cell surface and endosomal TLRS and intracellular pathways such as the NOD-like receptors, which confer responsiveness to viral nucleic acids, bacterial cell wall components and endotoxins. Once triggered via these pathways, keratinocytes release cytokines, such as type-I interferons, and chemo-attractants, including IL-8. Furthermore, keratinocytes contain a reservoir of pre-formed IL-1α, which they release upon activation, stimulating in an autocrine fashion and signalling to adjacent dermal fibroblasts in a paracrine fashion to promote wound healing responses and KGF release by the fibroblasts [22,48]. The importance of keratinocyte-derived IL-1α in epithelial–fibroblast cross-talk has been demonstrated in several papers [12,16]; moreover, in SSc, the feedback from dermal fibroblast synthesis of KGF has been confirmed as promoting the keratinocyte activation. 

Aberrant expression of both pro-inflammatory and pro-fibrotic factors has been demonstrated in and adjacent to the SSc epidermis. Demonstrated by profiling of candidate secreted factors, SSc epidermis synthesises the matricellular protein CCN2 (also called CTGF), which is found deposited at the epidermal–dermal interface [13]. Over-expression of CCN2 is thought to be a hallmark of fibrotic processes, acting as a modifier of cell–matrix interactions [49]. Though those studies look at the fibroblast response, other published data show a more pronounced presence of CCN2 in the epidermal–dermal junction in those with early SSc, suggesting that epidermal production of CCN2 could contribute to the pro-fibrotic response in SSc [13]. TGFβ is a major pro-fibrotic growth factor and is believed to have a role in SSc fibrosis [50]. However, when assayed, LAP-associated and total levels of TGFβ were not elevated in SSc epidermis [12]. Furthermore, the activated keratinocyte phenotype appears independent of this growth factor [29].

As mentioned above, IL-1α is considered a key initiator of keratinocyte activation during wound healing, and interestingly, IL1-α polymorphisms are known to be linked to increased risk and severity of SSc [51]. Moreover, the receptor IL-1R is overexpressed by SSc fibroblasts, and inhibition of IL-1R leads to the decreased release of IL-6 and PDGF-A in those fibroblasts, consistent with a paracrine pathway between keratinocytes and dermal fibroblasts [52]. In tissue culture models, SSc keratinocytes are able to stimulate fibroblasts through IL-1α release, leading to IL-6 and IL-8 release by fibroblasts [16]. Despite these preclinical data, when studied in patients, riloacapt, an IL-1 receptor biologic, failed to reduce skin score or decrease IL-6 levels in SSc patients [53].

Another potential pathway involving epithelial–fibroblast cross-talk is through S100A9 and its receptor TLR-4. S100A9 is a calcium-binding protein often present as a heterodimer with S100A8, released by activated myeloid cells as well as epithelial cells, promoting inflammatory cell recruitment and inducing cytokine release [54]. In fact, in SSc, overexpression of S100A9 has been confirmed in the epidermis, which is not seen in healthy controls. Moreover, when stimulated by S100A9, SSc fibroblasts oversecrete CCN2, the matricellular pro-fibrotic protein [13]. With the proliferation of single-cell RNAseq studies in SSc, overexpression of wound cytokeratins, as well as high levels of S100A8, have been confirmed in SSc keratinocytes [15].

Further insight into SSc pathogenesis is being derived from single-cell analysis of the lesional skin tissue. One such single-cell-based analysis was performed before and after JAK inhibitor (tofactinib) therapy given for diffuse SSc of less than 5 years duration. Focusing on the epithelial cells, the keratinocyte signature was confirmed as active in SSc, with K6-expressing populations and S100A8 populations identified. This study revealed significant enrichment for IFN signalling in basal, differentiating and keratinised cell layers, and profiling pre- and post-therapy indicated a downshift in type I and type II interferon signatures in basal and keratinised cell layers post-tocilizumab treatment [25]. However, single-cell approaches lack the spatial specificity required to unravel the nuances of aberrant keratinocyte function, making the current efforts at spatial transcriptomics an attractive further approach to studying this cell layer in this disease.

## 4. Cross-Talk and Transition Between Epithelial Cells and Mesenchymal Cells in SSc

Epithelial to mesenchymal transition (EMT) is a process of trans-differentiation in which epithelial cells lose polarity and switch to become mesenchymal cells, occurring during embryogenesis (Type I EMT), during tissue regeneration and fibrosis (Type 2) and during cancer invasion (Type 3) [55]. Biomarkers exist to identify the different subtypes of EMT. TGFβ, implicated in SSc fibrosis, is a known key driver of EMT: many studies on tissue cultures confirm that this factor can induce mesenchymal markers in epithelial cells [48,49].

In fact, in SSc tissues, epithelial cells do stain positive for FSP-1 and vimentin [14], and also demonstrate increased SNAI1, but not SNAI2, consistent with partially invoked EMT, without full commitment or invasion into the dermis. Additionally, data implicate that SFRP4, a Wnt modulator, is released by keratinocytes undergoing EMT [23,24]. Therefore, it is possible that cells undergoing EMT in the epidermis contribute to some of the pro-fibrotic tendency in SSc [51] in a process involving TFGβ and Wnt signalling.

A study by McCoy et al. was able to demonstrate further evidence for a pro-fibrotic cross-talk through increased expression of COL1A1 and α-SMA in healthy fibroblasts incubated with conditioned media from SSc keratinocytes [29].

As part of normal skin homeostasis, there is keratinocyte–fibroblast cross-talk maintaining a fully differentiated, functional epidermis with normal lipid content and skin permeability. Early work demonstrated a double paracrine mechanism involved in the regulation of tissue repair, where fibroblasts secrete KGF, which stimulates keratinocytes to produce 1L-α and this in turn stimulates fibroblast activation and production of KGF [56].

More recent work has shown that SSc fibroblasts show increased expression of KGF (likely through epigenetic mechanisms since it persists in tissue culture). It was demonstrated in SSc that fibroblast-derived KGF induces oncostatin M (OSM) production by keratinocytes, which in turn leads to OSM-mediated fibroblast activation through the phosphorylation of STAT3. This leads to increased collagen production and urokinase-type plasminogen activator (uPA), which enhances fibroblast migration [57]. This work further supports aberrancy in the epidermal–fibroblasts cross-talk in SSc, since when normal keratinocytes are co-cultured with normal fibroblasts, no increased STAT3-mediated activation of fibroblasts was seen.

## 5. Possible Initiating Biomechanisms Leading to Epidermis Activation in SSc

Whilst the evidence as presented, for abnormal activation of the keratinocyte layer in SSc is compelling, the initiating mechanisms remain to be determined. Possible explanations would include diffusion of growth factors and cytokines from the underlying disease dermis, effects of the mechano-stimulation from the enhanced ECM stiffness, or else autoimmune or environmental factors. Moreover, SSc IgG autoantibodies appear to bind and activate keratinocytes, leading to IL-1α release [58]. These mechanisms resemble the effects of anti-endothelial cell antibodies believed to have an initiating role in vascular damage in SSc [59]. It is possible that endothelial cell-initiating factors dominate in certain patients, whereas in others, epithelial cell activation is a key initiator (“inside to out“ fibrosis versus “outside to in” mechanism).

In idiopathic lung fibrosis, outside the setting of SSc, epithelial–fibroblast interactions are now established as being crucial to the fibrotic process. Environmental exposures leading to injured alveolar epithelial cells trigger myofibroblast activation and over-secretion of stiff ECM, creating a pro-fibrotic niche. It is possible that an “inside-out” type mechanism is occurring in some SSc patients through endothelial cell damage leading to paracrine fibrosis, whilst in others, an “outside-in” mechanism resulting from epithelial cell damage and epithelial–fibroblast cross-talk drives the disease process.

As alluded to already, the epidermis is also richly endowed with immune cells, including resident dendritic cells and transiting lymphocyte populations that could be initiating responses in this layer. Moreover, the effects of triggering injurious environmental factors such as industrial solvents might maximally induce innate responses in the epidermal cell layer [60].

## 6. Directing Therapies to the Epidermal Layer

Because of the multisystem, vascular and immune dysregulation, patients with severe forms of SSc are medically frail and susceptible to significant toxicities from immunomodulatory treatments. One possible approach is to cut down on systemic exposure by topical application of drugs, and in the case of restoring epidermal cell balance, the topical route is highly relevant. A number of strategies have evolved in the dermatology therapeutics field to selectively deliver drugs to this cell layer (reviewed in [61]). Possible approaches include drug delivery via transdermal patches, chemically enhanced skin-mediated delivery, and a range of techniques that rely on physical enhancement, such as microneedle patches, sonophoresis with ultrasound disruption of the epidermis, or iontophoresis, where an electrical current is used to deliver the drug into the epidermis. These methods aim to bypass or disrupt the natural keratinocyte barrier to facilitate drug absorption. These methods are generally painless, minimally invasive and less likely to cause systemic upset when compared to oral or subcutaneous routes. However, the potential for systemic absorption and significant systemic toxicity remains and requires full toxicity studies and pharmacokinetic analyses in any early phase study.

## 7. Future Directions

As the technology underlying biomedical research has progressed, ever more sophisticated approaches to understanding SSc as a disease process have become achievable. Latest profiling methodologies, such as single-cell, transcriptomics and proteomics, can be applied to the SSc epidermal cell layer in isolation, especially since it is readily available for biopsy or sampling. The high prevalence of gut and interstitial lung involvement could be explained by the importance of having a potentially activated or damaged epithelial cell layer.

Also, with the application of biologic therapies of known specific mode of action, tissues, including the epithelial layer, can be profiled before and after therapy to gain insights into the underlying molecular pathways involved, as illustrated in [62].

Finally, targeted treatment could be preferentially delivered into the epithelium by topical application, microneedle delivery or by inhaled nebulised route. Any subsequent improvement in the underlying deeper fibroblast-rich tissues could implicate the epithelia as a driving influence.

## Figures and Tables

**Figure 1 cells-14-00962-f001:**
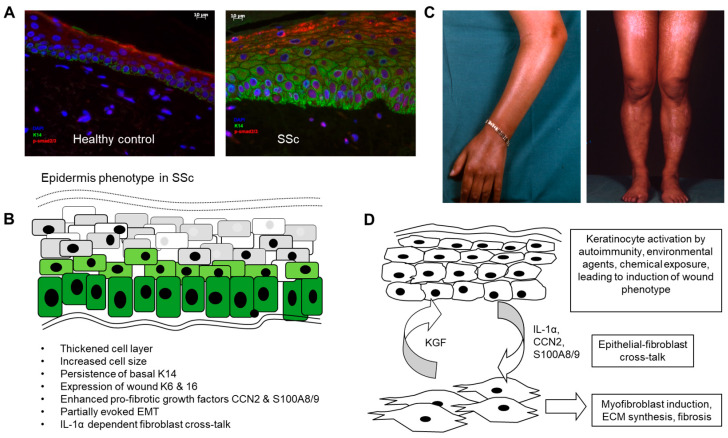
Role of the epidermis in systemic sclerosis. (**A**) Altered differentiation and activation of the epidermal cell layer in SSc. SSc epidermis was found to be thickened and to have abnormal persisting expression of cytokeratin 14 into suprabasal layers (green stain). pSMAD2/3, indicating active TGFβ signalling, was enhanced in the disease epidermis (red stain). (**B**) Summary of phenotypic changes described in various published articles and (**C**) pigmentary changes due to dysregulated melanosis in the epidermal layer. (**D**) Possible mechanisms linking epidermal activation to the underlying fibrotic process in SSc. Scale bar 10µm.

## Data Availability

Not applicable.

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
