# Peer review of "Understanding the Role of Epithelial Cells in the Pathogenesis of Systemic Sclerosis"

_cells, 2025, doi:10.3390/cells14130962_

Round 1
Reviewer 1 Report
Comments and Suggestions for Authors
In this study the authors discussed the role of epithelial cells in the pathogenesis of systemic sclerosis. Some concerns and suggestions are listed as below:
There are two 'Abstract: Abstract:' in line 7.
The English of this manuscript should be edited. For example, 'subset' should be 'subsets' in line 9.
Tables or figures can be provided to summary related findings (the role of epithelial cells in the pathogenesis of systemic sclerosis).
Latest profiling methodologies such as single cell, ATAC-seq and proteomics can be applied to the SSc epidermal cell layer in isolation, especially since it is readily available for biopsy or sampling. However, region-specific effects are not clear for readers. Other new methods such as spatial transcriptomics should also be discussed.
Targeted treatment could be preferentially delivered into the epithelial by topical 238
application, while potential side effects should not be ignored.
How about cross-talk between epithelial cells and other immune cells in the condition of SSc?
How to specifically target epithelial cells for the treatment of SSc?
Author Response
In this study the authors discussed the role of epithelial cells in the pathogenesis of systemic sclerosis. Some concerns and suggestions are listed as below:
There are two 'Abstract: Abstract:' in line 7.-thanks, now corrected
The English of this manuscript should be edited. For example, 'subset' should be 'subsets' in line 9.-now corrected, grammar and spell checked.
Tables or figures can be provided to summary related findings (the role of epithelial cells in the pathogenesis of systemic sclerosis).-main findings summarised in Figure 1 and addition of Table 1 summarising the phenotype changes.
Latest profiling methodologies such as single cell, ATAC-seq and proteomics can be applied to the SSc epidermal cell layer in isolation, especially since it is readily available for biopsy or sampling. However, region-specific effects are not clear for readers. Other new methods such as spatial transcriptomics should also be discussed.-now includes more detail about current findings of single cell analyses and potential to study the epidermal layer with transcriptomics
Targeted treatment could be preferentially delivered into the epithelial by topical 238
application, while potential side effects should not be ignored.-now discussed
How about cross-talk between epithelial cells and other immune cells in the condition of SSc?-section on immune cells in the epidermis has been added under Section 3
How to specifically target epithelial cells for the treatment of SSc?-new section 6 -directing therapies to the epidermal layer
In this review article, the authors emphasized the pathogenic role of epithelial/epidermal cells (interactions between epithelia and myofibroblasts in the dermal connective tissues) in systemic sclerosis (SSc) by referring the results in the previous studies by themselves and other authors. This is an interesting topic, which has not been well and deeply discussed so far. However, I have several comments which are described below.
(1) In the abstract section, the authors mainly described the background of this study and their speculation, but did not describe the results of the literature survey. Therefore, the authors should describe in more detail for their review results, specific to the pathogenic role of epithelia in SSc, particularly how epidermis affects to dermis and develop the symptoms of SSc.-thank you-whilst this was not intended as a meta-analysis or literature review, we now more extensively discuss the background literature and more from other groups. There is less personalised discussion of our own data now. I believe this has improved the overall balance of the article.
(2) This article describe about the effects from dermis to epidermis more extensively. However, the authors intended to show the pathogenic role of epidermis to dermis in the SSc pathology. Therefore, the authors should show more information about the pathogenic role of epidermis to dermis, which were reported in the previous studies.-thanks, now discussed and included.
(3) Figure 1c may not be cited in the text. "(Figure 1D)" appears earlier than "(Figure 1B)", which should be corrected.-now corrected-now only a single reference to (Figure 1).
(4) There are too many abbreviations, and some of them should be spelled out, particularly the abbreviations are used only once or twice in the text. In addition, the full spells should be shown in some abbreviations, such as "DAMPs", "PAMPs", "TLRS", "NOD", "KGF" and "LAP" at page 5.-OK thanks, les abbreviations used now, also now defined and corrected
Comments on the Quality of English Language
The structure and English of this manuscript need much improvements in many places.-thanks-now correctd for grammar and typographic errors
For example, space is inconsistently inserted before the citation of reference (i.e., before (11)) throughout the manuscript. This reviewer may suggest that a space may be inserted in all such place. Similarly, a space is missing at many places, where a space is needed. This should also carefully be corrected.-now corrected, thanks
"Abstract" is duplicated at the first place of the abstract section. The word "and" may be inserted between "autoantibodies" and "infiltration" at line 11. The sentence at lines 207-208 may be inappropriate.-thanks now corrected

Reviewer 2 Report
Comments and Suggestions for Authors
In this review article, the authors emphasized the pathogenic role of epithelial/epidermal cells (interactions between epithelia and myofibroblasts in the dermal connective tissues) in systemic sclerosis (SSc) by referring the results in the previous studies by themselves and other authors. This is an interesting topic, which has not been well and deeply discussed so far. However, I have several comments which are described below.
(1) In the abstract section, the authors mainly described the background of this study and their speculation, but did not describe the results of the literature survey. Therefore, the authors should describe in more detail for their review results, specific to the pathogenic role of epithelia in SSc, particularly how epidermis affects to dermis and develop the symptoms of SSc.
(2) This article describe about the effects from dermis to epidermis more extensively. However, the authors intended to show the pathogenic role of epidermis to dermis in the SSc pathology. Therefore, the authors should show more information about the pathogenic role of epidermis to dermis, which were reported in the previous studies.
(3) Figure 1c may not be cited in the text. "(Figure 1D)" appears earlier than "(Figure 1B)", which should be corrected.
(4) There are too many abbreviations, and some of them should be spelled out, particularly the abbreviations are used only once or twice in the text. In addition, the full spells should be shown in some abbreviations, such as "DAMPs", "PAMPs", "TLRS", "NOD", "KGF" and "LAP" at page 5.
Comments on the Quality of English LanguageThe structure and English of this manuscript need much improvements in many places.
For example, space is inconsistently inserted before the citation of reference (i.e., before (11)) throughout the manuscript. This reviewer may suggest that a space may be inserted in all such place. Similarly, a space is missing at many places, where a space is needed. This should also carefully be corrected.
"Abstract" is duplicated at the first place of the abstract section. The word "and" may be inserted between "autoantibodies" and "infiltration" at line 11. The sentence at lines 207-208 may be inappropriate.
Author Response

(The authors gave the same response as above.)

Round 2
Reviewer 1 Report
Comments and Suggestions for Authors
Previous concerns have been addressed by the authors.
Reviewer 2 Report
Comments and Suggestions for Authors
None.